# A Rate-Distortion Approach to Domain Generalization

## Abstract

Domain generalization deals with the difference in the distribution between the training and testing datasets, i.e., the domain shift problem. A principled approach to domain generalization is by extracting domain-invariant features. In this paper, we propose an information-theoretic approach for domain generalization. We first establish the domain transformation model, mapping a domain-free latent image into a domain. Then, we cast the domain generalization as a rate-distortion problem, and use the information bottleneck penalty to measure how well the domain-free latent image is reconstructed from a compressed representation of a domain-specific image compared to its direct prediction from the domain-specific image itself. We prove that the information bottleneck penalty guarantees that domain-invariant features can be learned. Lastly, we draw links of our proposed method with self-supervised contrastive learning without negative data pairs. Our empirical study on two different tasks verifies the improvement over recent baselines.

## 1 Introduction

Deep neural networks (DNNs) are highly expressive models that reach state-of-the-art performance in challenging tasks, such as speech and visual recognition (Devlin et al., 2018; He et al., 2016), by capturing complex correlations among input elements, e.g., pixels of an image. However, the correlations might also contain spurious features that hurt the generalization performance of DNNs on out-of-distribution samples (Szegedy et al., 2013; Beery et al., 2018; Alcorn et al., 2019). Unfortunately, real-world applications often encounter such out-of-distribution samples, e.g., when the training domain does not match the testing domain. A prominent example is deblurring, where models are trained on simulated blurs which differ substantially to real-world blurring (Koh et al., 2021). In other words, generalization across domains is a critical task before deploying DNNs to real-world application.

Learning features that are invariant across multiple training domains, and using those features for out-of-distribution generalization has emerged as a significant topic in domain generalization. In domain generalization, multiple source domains are accessible during training, but the target domains are not (Blanchard et al., 2011; Muandet et al., 2013). Invariant risk minimization (IRM) (Arjovsky et al., 2019) is a prominent approach for learning domain invariant features. However, IRM suffers from the case when the invariant features contains full information about the label (Ahuja et al., 2020). To deal with this shortcoming, Ahuja et al. (2021) introduce the information bottleneck theory on neural networks (Tishby & Zaslavsky, 2015), and show that their method will be guaranteed to converge to the invariant features. On the empirical side, a series of works align source domain distributions for domain-invariant representation learning by either direct construct auxiliary penalty (Duan et al., 2012; Sun & Saenko, 2016; Li et al., 2018b;c; 2017; Niu et al., 2015), or meta learning (Li et al., 2019; Balaji et al., 2018; Li et al., 2018a).

There are also series of work do not rely on invariant features. They can be categorized as (1) domain-specific method: Domain2Vec (D2V) (Deshmukh et al., 2018) learns domain-specific embedding, DMG (Chattopadhyay et al., 2020) aims to learn domain specific masks; and (2) augmentation method: (Volpi et al., 2018) augments the dataset adversarially, L2A-OT (Zhou et al., 2020) augments data with image information. Despite their success, there is no guarantee that empirical methods can solve the task across different environments.

In this paper, we use an information-theoretic approach to tackle domain generalization. We assume there is a domain-free latent instance (e.g., an image) that captures the invariant features we want to extract. We define a domain transformation model that maps the domain-free latent instance into a domain and then we apply the rate distortion theory to obtain a domain-invariant representation. The proposed method, called Twins, is guaranteed to converge to the invariant feature under the linear classification structural equation model (Ahuja et al., 2021). We evaluate our method on linear unit tests (Aubin et al., 2021) and variants of MNIST dataset (LeCun & Cortes, 2010; Xiao et al., 2017; Clanuwat et al., 2018), which validates the theoretical analysis and demonstrates how the proposed method can outperform the previous ones. Our contributions can be summarized as follows:

- We cast domain generalization as a rate distortion problem and prove how the proposed method can converge.
- We illustrate how the proposed method extends previous results on domain generalization, and draw links to self-supervised contrastive learning. We demystify the success of contrastive learning by giving a contrastive learning based domain generalization algorithm with theoretical guarantee.
- We evaluate our method on two datasets and observe consistent improvement over existing baselines.

## 2 PRELIMINARY ON DOMAIN GENERALIZATION

Assume that the instance-label pair $(X, Y)$ is sampled from an unknown distribution $\mathbb{P}(X, Y)$. The objective of standard supervised learning is to learn a predictor $f$ that is able to predict the labels $Y$ of corresponding instances $X$ for each $(X, Y) \sim \mathbb{P}(X, Y)$, given the finite training samples drawn from the underlying distribution $\mathbb{P}(X, Y)$.

Unlike the standard supervised learning tasks, in domain generalization, we cannot sample directly from the distribution $\mathbb{P}(X, Y)$. Instead, we can only observe $(X, Y)$ under different domains $e \in \mathcal{E}_{\text{all}}$, denoted as $(X^e, Y^e) \sim \mathbb{P}^e(X^e, Y^e)$. We also assume that $e \in \mathcal{E}_{\text{all}}$ is distributed as $e \sim \mathbb{P}_e$. Given samples from a finite subset $\mathcal{E}_{\text{train}} \subsetneq \mathcal{E}_{\text{all}}$ of all the domains, the goal of the domain generalization problem is to learn a predictor $f$ that generalizes across all possible domains. This can be summarized as follows:

**Problem 2.1** (Domain generalization). *Let $\mathcal{E}_{train} \subsetneq \mathcal{E}_{all}$ be a finite subset of training domains. We have access to the data for each training domain $e_{train} \in \mathcal{E}_{train}$, but have no access to the data for each test domain $e_{test} \in \mathcal{E}_{all} \backslash \mathcal{E}_{train}$. Given a function class $\mathcal{F}$ and a loss function $\ell$, our goal is to learn a predictor $f \in \mathcal{F}$ using the data from the training domain such that $f$ minimizes the worst-case risk over $\mathcal{E}_{all}$. Define the risk of the predictor $f$ on the domain $e$ as $R^e(f) := \mathbb{E}_{\mathbb{P}^e(X^e, Y^e)} \ell(f(X^e), Y^e)$. We want to solve the following min-max optimization problem:*

$$\underset{f \in \mathcal{F}}{\text{minimize}} \ \max_{e \in \mathcal{E}_{all}} R^e(f). \tag{DG}$$

We establish the domain transformation model to characterize the relation between domain-aware instance $X^e$ and the domain-invariant latent instance $X$ in the Assumption 1, which first appears in Robey et al. (2021).

**Assumption 1** (Domain transformation model). *Let $\delta_e$ denote a Dirac distribution for $e \in \mathcal{E}_{all}$. We assume that there exists a measurable function $G : \mathcal{X} \times \mathcal{E}_{all} \to \mathcal{X}$, which we refer to as a* domain transformation model*, that parameterizes the inter-domain covariate shift via*

$$\mathbb{P}^e(X) =^d G \# (\mathbb{P}(X) \times \delta_e) \quad \forall e \in \mathcal{E}_{all}, \tag{1}$$

*where $\#$ denotes the push-forward measure and $=^d$ denotes equality in distribution.*

The Assumption 1 can somewhat reflect the generation of domain specific instances. For example, the multiple different views of a 3D object (Niu et al., 2015), different angles of the image (Rotated MNIST (Worrall et al., 2017)). Besides, the MUNIT architecture (Huang et al., 2018) can effectively distangle the domain-free latent instance $X$ and the specific environment $e$, and thus can be used as the domain transformation model $G$ (Robey et al., 2021).

Let $\Phi$ denote the feature representation mapping, $w$ denote the classifier and $w \circ \Phi$ denote the full predictor. The regret of the network on the domain $e$ is denoted as $R^e(w \circ \Phi)$.

Next, we define standard properties related to the datasets used in the domain generalization literature (Ahuja et al., 2021). For each $e \in \mathcal{E}_{all}$, the distribution $(X^e, Y^e) \sim \mathbb{P}^e$ satisfies the following properties: (1) $\exists$ a map $\Phi^*$, which we call an *invariant feature map*, such that $\mathbb{E}\left[Y^e \middle| \Phi^*(X^e)\right]$ is the same for all $e \in \mathcal{E}_{all}$ and $Y^e \not\perp \Phi^*(X^e)$, where $\perp$ means mutual independence. (2) $\exists$ a map $\Psi^*$, which we call *spurious feature map*, such that $\mathbb{E}\left[Y^e \middle| \Psi^*(X^e)\right]$ is not the same for all $e \in \mathcal{E}_{all}$ and $Y^e \not\perp \Psi^*(X^e)$ for some domains. $\Psi^*$ often hinders learning predictors that only rely on $\Phi^*$. For example, in the CMNIST dataset, the $\Phi^\star$ extracts the underlying digit and $\Psi^\star$ extracts background color.

The baseline algorithm for domain generalization Equation (DG) is the Empirical Risk Minimization, i.e. directly minimizing the empirical risk on the training domains:

$$\min_{w,\Phi} \frac{1}{|\mathcal{E}_{\text{train}}|} \sum_{e \in \mathcal{E}_{\text{train}}} R^e(w \circ \Phi), \tag{2}$$

where $|\mathcal{E}_{\text{train}}|$ denotes the number of training domains.

We say that a data representation $\Phi$ elicits an invariant predictor across the set of training domains $\mathcal{E}_{\text{train}}$ if there is a predictor $w$ that simultaneously achieves the minimum risk, i.e. $w \in \arg\min_{w'} R^e(w' \circ \Phi), \forall e \in \mathcal{E}_{\text{train}}$. Using this notation, the main objective of Invariant Risk Minimization (IRM) is stated as:

$$\min_{w,\Phi} \frac{1}{|\mathcal{E}_{\text{train}}|} \sum_{e \in \mathcal{E}_{\text{train}}} R^e(w \circ \Phi), \qquad \text{s.t.} \ \ w \in \arg\min_{w'} R^e(w' \circ \Phi), \ \forall e \in \mathcal{E}_{\text{train}}. \tag{3}$$

Lastly, we rely on the notion of 'informativeness' about the datasets (Ahuja et al., 2021). There are two such categories of informativeness. In the first case, the invariant features $\Phi^*(X^e)$ are *partially informative* about the label, i.e. $Y \not\perp X^e | \Phi^*(X^e)$, and color contains information about label not contained in the uncolored digit. In the second case, invariant features are *fully informative* about the label, i.e., $Y \perp X^e | \Phi^*(X^e)$, i.e., they contain all the information about the label that is contained in input $X^e$. Many real-world image datasets have fully informative invariant features, the labels are a deterministic function of the domain-invariant features and domain-aware spurious features do not affect the label.

## 3 METHOD

### 3.1 RATE DISTORTION & INFORMATION BOTTLENECK PRINCIPLE

Given a domain-free latent instance $X \in \mathbb{R}^{d_X}$, its observation in a domain $e$ is denoted as $X^e := G(X, e)$. We want to learn the feature $Z^e = \Phi(X^e) \in \mathbb{R}^{d_Z}$ which is informative about the domain-free variable $X$, but invariant (i.e. uninformative) to the specific domain $e$. We use rate–distortion theory (Davisson, 1972; Blau & Michaeli, 2019) to formulate our domain generalization problem.

Rate–distortion theory is a major branch of information theory which provides the theoretical foundations for lossy data compression. An encoder $\Phi$ encodes domain-aware instances $X^e$. We want the representation $Z^e = \Phi(X^e)$ to be domain-invariant, so we feed $Z^e$ into a decoder which outputs domain-invariant $X$. We minimize the distortion between the original domain-aware instance $X^e$ and the reconstructed domain-free instance $X$. The distortion function measures how well $X$ is predicted from a compressed representation $Z^e$ compared to its direct prediction from $X^e$. This trade-off is captured by the following loss function:

$$L_{IB}(\theta, e) = \mathbb{E}_{X \sim \mathbb{P}_X, e \sim \mathbb{P}_e} I(Z^e; X^e) - \beta I(Z^e; X) \quad \text{IB objective} \tag{4}$$

where $I$ denotes the mutual information, $\theta$ is the parameter of the representation function $\Phi$, and $\beta$ is a constant.

In the following, we consider two cases: discrete and continuous variables, owing to their different definition of entropy.

**Discrete case**: Since the representation function is deterministic with respect to $\theta$, we can rewrite Equation (4) through a classical identity of mutual information: $I(X; Y) = H(X) - H(X|Y)$,

where $H$ denotes the Shannon entropy for discrete variables, as follows:

$$
\begin{aligned}
L_{IB}(\theta, e) &= \mathbb{E}_{X,e} I(Z^e; X^e) - \beta I(Z^e; X) \\
&= \mathbb{E}_{X,e} H(Z^e) - H(Z^e|X^e) - \beta(H(Z^e) - H(Z^e|X)) \\
&= \mathbb{E}_{X,e} H(Z^e|X) + \frac{1-\beta}{\beta} H(Z^e),
\end{aligned}
\tag{5}
$$

where in the last equality we omit the overall scaling factor of the loss function.

If $0 \leq \beta \leq 1$, since $H(\cdot)$ is bounded below by 0, setting $\Phi$ to be constant will clearly minimize the penalty, which is uninformative about the representations we want to learn. Hence, we set $\beta > 1$, and replace $\frac{1-\beta}{\beta}$ with $-\lambda$, where $0 \leq \lambda < 1$. The IB objective can be rewritten as

$$
L_{IB}(\theta, e) = \mathbb{E}_{X,e} H(Z^e|X) - \lambda H(Z^e).
\tag{6}
$$

**Continuous case**: In terms of continuous variables, the differential entropy $h(\cdot)$ is not bounded below, which hinders our analysis. To overcome this, we can define the lower bounded differential entropy $\widehat{h}(X) := h(X + \varepsilon)$, where $\varepsilon$ is the independent bounded zero-entropy noise $\varepsilon \sim \mathrm{Uniform}(0, 1)$. Thus, $\widehat{h}(X) \geq h(\varepsilon) = 0$. We can replace the Shannon entropy $H(\cdot)$ with lower bounded differential entropy $\widehat{h}(\cdot)$ in Equation (6):

$$
L_{IB}(\theta, e) = \mathbb{E}_{X,e} \widehat{h}(Z^e|X) - \lambda \widehat{h}(Z^e).
\tag{7}
$$

For simplicity, we define $H$ to be the Shannon entropy for the discrete variables, or lower bounded entropy $\widehat{h}$ for continuous variables in the main text. We define $H^e(f) := \mathbb{E}_{X^e \sim \mathbb{P}^e} H(f(X^e))$. We can extend the Empirical Risk Minimization (ERM) algorithm to include the IB Penalty, and the resulting algorithm, denoted as Twins-ERM method, is the following:

$$
\min_{w,\Phi} \sum_{e \in \mathcal{E}_{\text{train}}} H^e(\Phi|X) - \lambda H^e(\Phi) \quad \text{s.t.} \quad \frac{1}{|\mathcal{E}_{\text{train}}|} \sum_{e \in \mathcal{E}_{\text{train}}} R^e(w \circ \Phi) \leq r
\tag{8}
$$

where $r$ is the threshold on the empirical risk on the training domains.

In addition to ERM, another popular minimization framework is the invariant risk minimization (Arjovsky et al., 2019). The proposed penalty can be readily incorporated into the IRM framework, we call the resulting algorithm Twins-IRM:

$$
\begin{aligned}
&\min_{w,\Phi} \sum_{e \in \mathcal{E}_{\text{train}}} H^e(\Phi|X) - \lambda H^e(\Phi), \\
&\text{s.t.} \ \frac{1}{|\mathcal{E}_{\text{train}}|} \sum_{e \in \mathcal{E}_{\text{train}}} R^e(w \circ \Phi) \leq r, \ w \in \arg\min_{\tilde{w}} R^e(\tilde{w} \circ \Phi), \forall e \in \mathcal{E}_{\text{train}}.
\end{aligned}
\tag{9}
$$

### 3.2 Theoretical Guarantee

In this subsection, we establish the theoretical guarantee of our algorithm under the linear classification case. We consider the following standard 0-1 classification model in literature (Ahuja et al., 2020; 2021):

**Assumption 2.** *Linear classification structural equation model. In each $e \in \mathcal{E}_{all}$,*

$$
\begin{aligned}
Y^e &\leftarrow \mathrm{I}(w_{\text{inv}}^\star \cdot X_{\text{inv}}) \oplus N^e, \quad N^e \sim \mathrm{Bernoulli}(q), q \leq \frac{1}{2}, \quad N^e \perp (X_{\text{inv}}, X_{\text{spu}}^e) \\
X^e &\leftarrow S(X_{\text{inv}}^e, X_{\text{spu}}^e), \quad X_{\text{inv}}^e \leftarrow G(X_{\text{inv}}, e)
\end{aligned}
\tag{10}
$$

*where $\mathrm{I}(x)$ is 1 if $x$ is positive else 0, $w_{\text{inv}}^\star \in \mathbb{R}^m$ with $\|w_{\text{inv}}^\star\| = 1$ is the labelling hyperplane, $X_{\text{inv}}^e, X_{\text{inv}}^e \in \mathbb{R}^m, X_{\text{spu}}^e \in \mathbb{R}^o, S \in \mathbb{R}^{(m+o) \times (m+o)}$ and $G$ is a continuous domain transformation.*

Before presenting our main theorem, we first add two assumptions on the support of invariant features. Define the support of the invariant features $X_{\text{inv}}^e$ in environment $e$ as $\mathcal{X}_{\text{inv}}^e$.

**Assumption 3** (Invariant feature support overlap). *The union of support of the invariant features of the training domains covers support of the invariant features of all the domains. i.e. $\bigcup_{e \in \mathcal{E}_{all}} \mathcal{X}^e_{\text{inv}} \subseteq \bigcup_{e \in \mathcal{E}_{train}} \mathcal{X}^e_{\text{inv}}$.*

**Assumption 4** (Strictly separable invariant features). *The training support of invariant features $\bigcup_{e \in \mathcal{E}_{train}} \mathcal{X}^e_{\text{inv}}$ is strictly separated by the labelling hyperplane $w^*_{\text{inv}}$. In other words, $\min_{x \in \bigcup_{e \in \mathcal{E}_{train}} \mathcal{X}^e_{\text{inv}}} \text{sign}(w^*_{\text{inv}} \cdot x) \cdot (w^*_{\text{inv}} \cdot x) > 0$.*

Assumption 3 and 4 describes the property of the support of invariant feature. Under these assumptions, we propose our first main theorem:

**Theorem 3.1.** *Suppose each $e \in \mathcal{E}_{all}$ follows Assumption 2, and Assumptions 3 and 4 hold for the invariant features. Also, for each $e \in \mathcal{E}_{train}$, assume that $X^e_{\text{spu}} = AX^e_{\text{inv}} + W^e$, where $A \in \mathbb{R}^{o \times m}$, $W^e \in \mathbb{R}^o$ is continuous, bounded, and zero mean noise. Each solution to Twins-ERM and Twins-IRM (Equation (8) and Equation (9), with $\ell$ as 0-1 loss, and $r = q$) solves the domain generalization problem (Equation (DG)).*

*Sketch of Proof.* The full proof is provided at Appendix A. We only present the main idea here. Denote $\Phi^\dagger$ as the solution to Equation (9),

$$\Phi^\dagger X^e = \Phi^\dagger S(X^e_{\text{inv}}, X^e_{\text{spu}}) = \Phi^\dagger_{\text{inv}} X^e_{\text{inv}} + \Phi^\dagger_{\text{spu}} X^e_{\text{spu}} = (\Phi^\dagger_{\text{inv}} + \Phi^\dagger_{\text{spu}} \cdot A) X^e_{\text{inv}} + \Phi^\dagger_{\text{spu}} W^e \quad (11)$$

We will show that $\Phi^+ = \left( \left[ \Phi_{\text{inv}} + \Phi_{\text{spu}} \cdot A \right], 0 \right) S^{-1}$ can continue to achieve an error of $q(= r)$ across training domains, and have a lower information bottleneck penalty. Therefore, the optimal solution to Twins-ERM (Equation (8)) does not depend on the spurious noise $W^e$, and hence solves the domain generalization problem (Equation (DG)). $\square$

It is known that ERM and IRM fails under the assumption of Theorem 3.1 (Theorem 3 in (Ahuja et al., 2021)). This theorem shows that our algorithm can provably solve the linear classification structural equation model.

In real world, however, we do not have direct access to the domain-free instance $X$. Hence, we practically adopt image from another domain, denoted as $X^{e'}$, as a proxy for $X$ in Twins-ERM and Twins-IRM (Equation (8) and Equation (9)). In other words, fixing $e' \in \mathcal{E}_{\text{train}}$, the Twins-ERM (Equation (8)) can be rewritten as

$$\min_{w, \Phi} \sum_{e \in \mathcal{E}_{\text{train}}} H^e\big(\Phi(X^e)|X^{e'}\big) - \lambda H^e(\Phi) \quad \text{s.t.} \quad \frac{1}{|\mathcal{E}_{\text{train}}|} \sum_{e \in \mathcal{E}_{\text{train}}} R^e(w \circ \Phi) \leq r, \quad (12)$$

and the Twins-IRM (Equation (9)) can be rewritten as

$$\min_{w, \Phi} \sum_{e \in \mathcal{E}_{\text{train}}} H^e\big(\Phi(X^e)|X^{e'}\big) - \lambda H^e(\Phi)$$
$$\text{s.t.} \quad \frac{1}{|\mathcal{E}_{\text{train}}|} \sum_{e \in \mathcal{E}_{\text{train}}} R^e(w \circ \Phi) \leq r, , \; w \in \arg\min_{\tilde{w}} R^e(\tilde{w} \circ \Phi), \forall e \in \mathcal{E}_{\text{train}}. \quad (13)$$

We will show adopting proxy from another domain will still be guaranteed to solve the domain generalization problem (Equation (DG)).

**Theorem 3.2.** . *Suppose each $e \in \mathcal{E}_{all}$ follows Assumption 2, and Assumptions 3 and 4 hold for the invariant features. Also, for each $e \in \mathcal{E}_{train}$, assume that $X^e_{\text{spu}} = AX^e_{\text{inv}} + W^e$, where $A \in \mathbb{R}^{o \times m}$, $W^e \in \mathbb{R}^o$ is continuous, bounded, and zero mean noise. Each solution to Twins-ERM and Twins-IRM (Equation (12) and Equation (13)), with $\ell$ as 0-1 loss, and $r = q$) solves the domain generalization problem (Equation (DG)).*

The full proof of Theorem 3.1 and 3.2 can be found at Appendix A.

### 3.3 GAUSSIAN BOTTLENECK

The IB obejctive (4) can be directly estimated by k-nearest-neighbor method, as described in Appendix C. However, such direct estimation requires large memory and is computationally intensive.

We denote the algorithm by kNN direct estimation as Twins-Direct. In order to facilitate the implementation of our penalty, we make the simplifying assumption that the datasets follow a Gaussian distribution (Chechik et al., 2005). Specifically, assuming $X, X^e$ are jointly multivariate zero-mean Gaussian vectors with covariances $\Sigma_X, \Sigma_{X^e}$, and $Z^e \in \mathbb{R}^{d_Z}$ is a encoded version of $X^e$ that must maintain a given value of mutual information with $X$. We define the covariance of $Z^e$ and $Z^e|X$ to be $\Sigma_{Z^e}$ and $\Sigma_{Z^e|X}$.

Next, we simplify the Equation (6). The entropy of a Gaussian distribution is simply given by the logarithm of the determinant of its covariance function (up to a constant that we ignore). The loss function becomes:

$$L_{IB}(\theta, e) = \mathbb{E}_X \log \det(\Sigma_{Z^e|X}) - \lambda \log \det(\Sigma_{Z^e}). \tag{14}$$

**Practical considerations**: We reformulate our algorithm so that it resembles the contrastive learning method, i.e. Barlow Twins (Zbontar et al., 2021). The second term of the loss in Equation (14) maximizes $\det(\Sigma_{Z^e})$. Since computing the determinant of a matrix is computationally intensive, we adopt a proxy to minimize the Frobenius norm of the correlation matrix of $Z^e$. Since the correlation matrix is invariant to scaling, we can set the diagonal element of the correlation matrix of $Z^e$ to be 1. Then, the second term of Equation (14) amounts to minimizing the off-diagonal term, i.e. the second term in Equation (16). This term essentially decorrelates the different dimensions of the representation and prevents these dimensions from encoding similar information.

Besides, it can also easily be shown that the first term of Equation (14) minimizes the information the representation contains about the domain information has the same solution with the first term of Equation (16). This term maximizes the alignment between representations of pairs of domain-aware instances $X^e$ and domain-free instances $X$.

In practice, we have *no access to the domain-free latent instance*. For a given instance $X^e$, we use an instance with the same label, but from a different domain, denoted as $X^{e'}$ as surrogate for the domain-free latent $X$. We minimize the distance between pairs of instances from different domains. Sample $\{z_b^e\}_{1 \le b \le B} \sim Z^e$ and $\{z_b^{e'}\}_{1 \le b \le B} \sim Z^{e'}$, we concatenate them into matrix $\mathbf{Z}^e \in \mathbb{R}^{B \times d_Z}$ and $\mathbf{Z}^{e'} \in \mathbb{R}^{B \times d_Z}$, where $d_Z$ is the dimension of $z_b^e, z_b^{e'}$. After mean shifting every column of $\mathbf{Z}^e$ and $\mathbf{Z}^{e'}$, such that $\mathbb{1}^\top \mathbf{Z}^e = \mathbb{1}^\top \mathbf{Z}^{e'} = 0$ ($\mathbb{1}$ is a column vector full of 1s), the cross-correlation matrix $C_{ij}^Z$ is defined as:

$$C_{ij}^Z = \frac{\langle \mathbf{Z}_{:,i}^e, \mathbf{Z}_{:,j}^{e'} \rangle}{\|\mathbf{Z}_{:,i}^e\|_2 \|\mathbf{Z}_{:,j}^{e'}\|_2}, 1 \le i, j \le d_Z, \tag{15}$$

and the final penalty is defined as:

$$c(Z) = \sum_i (1 - C_{ii}^Z)^2 + \lambda \sum_{i \ne j} (C_{ij}^Z)^2. \tag{16}$$

However, we do not have access to domain transformation model either. We construct the contrastive instances by permuting the instances that have the same label in each iteration. In particular, we sample $B$ instances $\{x_b\}_{1 \le b \le B}$ from training domains as row vectors, where $B$ is the batch size. We concatenate the representation $\{z_b\} = \{\Phi(x_b)\}$ into matrix $\mathbf{Z}^1 \in \mathbb{R}^{B \times d_Z}$. The contrastive batch $\mathbf{Z}^2$ is constructed by permuting the rows of $\mathbf{Z}^1$, i.e. $Z_{b,:}^1 = Z_{\pi(b),:}^2$, where $\pi$ is a permutation of $\{1, 2, \cdots, B\}$ such that the corresponding labels of $x_b$ and $x_{\pi(b)}$ are identical. $C_{ij}^Z$ defined in Equation (15) can be rewritten as:

$$C_{ij}^Z = \frac{\langle \mathbf{Z}_{:,i}^1, \mathbf{Z}_{:,j}^2 \rangle}{\|\mathbf{Z}_{:,i}^1\|_2 \|\mathbf{Z}_{:,j}^2\|_2}, 1 \le i, j \le d_Z \tag{17}$$

Such penalty can be readily incorporated into the ERM and IRM losses, i.e.

$$L_{\text{Twins-ERM}} = L_{\text{ERM}} + \mu \cdot c(Z), \text{ and } L_{\text{Twins-IRM}} = L_{\text{IRM}} + \mu \cdot c(Z), \tag{18}$$

where $L_{\text{ERM}}$ and $L_{\text{IRM}}$ denote the loss in the ERM and IRM respectively, and $\mu$ is the penalty hyperparameter.

## 4 EXPERIMENTS

In this section, we conduct experimentation on two benchmarks: Linear Unit Tests (in Section 4.1, Section 4.3) and DomainBed (in Section 4.2). Linear Unit Tests (Aubin et al., 2021) consists of several toy datasets to evaluate algorithms for domain generalization and invariance learning, while DomainBed (Gulrajani & Lopez-Paz, 2020) is a unified testbed for evaluating domain generalization algorithms. We use the following *four baselines* across our experiments: ERM, IB-ERM, IRM, IB-IRM (Ahuja et al., 2021) in the synthetic data, and use ERM, IRM as *baselines* in real-world datasets.

### 4.1 LINEAR UNIT TESTS

The dataset describes six linear low-dimensional problems, named Example 1/1s, Example 2/2s and Example 3/3s, where the 's' dictates a different rotation matrix. Each example, called unit test, is designed to test different types of out-of-distribution generalization. We describe in Appendix B.1 the precise distributions and the invariances captured by each example.

**Benchmark details**: We follow the same pipeline as those used in Aubin et al. (2021); Ahuja et al. (2021) for the model selection, hyperparameter selection, training, and evaluation. We set $(d_{\mathrm{inv}}, d_{\mathrm{spu}}) = (5, 5)$. For all three examples, the models used are linear. The training loss is the square error for the regression setting (Example 1/1s), and binary cross-entropy for the classification setting (Example 2/2s, 3/3s). For the evaluation of performance on Example 1/1s, we report mean square errors and standard deviations. For the evaluation of performance on Example 2/2s, Example 3/3s, we report classification errors and standard deviations.

**Model training**: For the Twins-ERM approach, there is an additional hyperparameter $\mu$ associated with the $c(Z)$ term in the final objective in Equation (18). We sample the $\mu$ from $\log \mu \sim \mathrm{Uniform}(-3, -1)$. For each algorithm, we run a random hyperparameter search for 20 trials, and average the results over 50 data seeds. We train each algorithm and hyperparameter trial on the train splits of all environments, for $10^4$ full-batch Adam updates (Kingma & Ba, 2014). We choose the hyperparameters trial that minimizes the error on the validation splits of all environments, i.e. the train-domain validation set evaluation procedure in (Gulrajani & Lopez-Paz, 2020).

We also implement an Oracle that contains randomized $X_{\mathrm{spu}}$ in each iteration, such that it learns to ignore the spurious features.

| | #Envs | ERM | IB-ERM | IRM | IB-IRM | Twins-ERM | Twins-IRM | Oracle |
|---|---|---|---|---|---|---|---|---|
| Example1 | 3 | $13.36 \pm 1.49$ | $12.96 \pm 1.30$ | $\mathbf{11.15 \pm 0.71}$ | $11.68 \pm 0.90$ | $14.62 \pm 1.20$ | $14.42 \pm 0.86$ | $10.42 \pm 0.16$ |
| Example1s | 3 | $13.33 \pm 1.49$ | $12.92 \pm 1.30$ | $\mathbf{11.07 \pm 0.68}$ | $11.74 \pm 1.03$ | $14.64 \pm 1.22$ | $13.25 \pm 1.49$ | $10.45 \pm 0.19$ |
| Example2 | 3 | $0.42 \pm 0.01$ | $0.00 \pm 0.00$ | $0.45 \pm 0.00$ | $0.00 \pm 0.00$ | $\mathbf{0.00 \pm 0.00}$ | $0.00 \pm 0.00$ | $0.00 \pm 0.00$ |
| Example2s | 3 | $0.45 \pm 0.01$ | $0.00 \pm 0.01$ | $0.45 \pm 0.00$ | $0.06 \pm 0.12$ | $\mathbf{0.00 \pm 0.00}$ | $0.43 \pm 0.03$ | $0.00 \pm 0.00$ |
| Example3 | 3 | $0.48 \pm 0.07$ | $0.49 \pm 0.06$ | $0.48 \pm 0.07$ | $0.48 \pm 0.07$ | $\mathbf{0.42 \pm 0.15}$ | $0.33 \pm 0.14$ | $0.00 \pm 0.00$ |
| Example3s | 3 | $0.49 \pm 0.06$ | $0.49 \pm 0.06$ | $0.49 \pm 0.07$ | $0.49 \pm 0.07$ | $0.50 \pm 0.05$ | $\mathbf{0.42 \pm 0.11}$ | $0.00 \pm 0.00$ |
| Example2 | 6 | $0.37 \pm 0.06$ | $0.02 \pm 0.05$ | $0.46 \pm 0.01$ | $0.43 \pm 0.11$ | $\mathbf{0.00 \pm 0.00}$ | $0.07 \pm 0.11$ | $0.00 \pm 0.00$ |
| Example2s | 6 | $0.46 \pm 0.01$ | $0.02 \pm 0.06$ | $0.46 \pm 0.01$ | $0.45 \pm 0.10$ | $\mathbf{0.00 \pm 0.00}$ | $0.47 \pm 0.00$ | $0.00 \pm 0.00$ |
| Example3 | 6 | $0.33 \pm 0.18$ | $0.26 \pm 0.20$ | $\mathbf{0.14 \pm 0.18}$ | $0.19 \pm 0.19$ | $0.24 \pm 0.16$ | $0.25 \pm 0.20$ | $0.01 \pm 0.00$ |
| Example3s | 6 | $0.36 \pm 0.19$ | $0.27 \pm 0.20$ | $\mathbf{0.14 \pm 0.18}$ | $0.19 \pm 0.19$ | $0.31 \pm 0.19$ | $0.44 \pm 0.06$ | $0.01 \pm 0.00$ |

Table 1: Comparisons on linear unit tests in terms of mean square error (regression, ↓) in Example 1/1s and classification error (classification, ↓) in Examples 2/2s and 3/3s. The highlighted result per example demonstrates the best performance. When #Envs=6, we do not report results on Example 1/1s, since even the oracle cannot obtain stable results across different data seeds.

The experimental results are reported in Table 1. In the Example 2/2s, since the invariant feature contains full information about the label, we do observe that IB penalty in (Ahuja et al., 2021) and Twins penalty in our paper performs the best. In the Example 1/1s and 3/3s, the spurious feature contains partial information about the label, we generally find invariant risk can reduce the error in this case. We empirically verify the benefit of using the proposed penalty over the previously proposed baselines.

### 4.2 MNIST-TYPE DATASET

In the second benchmark we use DomainBed to experiment on MNIST-type datasets inspired by the construction of CS-CMNIST (Ahuja et al., 2021) to evaluate covariate shift. In addition to the

origin MNIST (LeCun & Cortes, 2010), we extend the benchmark to include FashionMNIST (Xiao et al., 2017) and KMNIST (Clanuwat et al., 2018), where the images of the latter two are used as drop-in replacements for MNIST images. The idea in CS-CMNIST is to associate each class with a color, and each image is assigned the color associated to its class with probability $p^e$ or a random color with probability $1 - p^e$. We construct three environments for this experiment: two training environments containing 20,000 data points each, one test containing 20,000 points. In the two training environments, the $p^e$ is set to 1.0 and 0.9 respectively. In the testing environment, the $p^e$ is set to 0, i.e., all the images are colored completely at random. A grid search in the range of $\{10^{-4}, 10^{-3}, 10^{-2}, 10^{-1}, 1\}$ is used to determine the optimal penalty parameter $\mu$. We fix the trade-off parameter $\lambda = 5 \times 10^{-3}$. We run the experiments using 5 different seeds and report the mean and the standard deviation of the classification.

The results are reported in Table 2. We find that generally setting $\mu = 10^{-2}$ would be the best choice. See Appendix D Notice that both the IRM and the ERM versions in each case perform similarly. The results reveal that ERM and IRM have the weakest performance. IB-ERM and IB-IRM increase the accuracy of ERM and IRM respectively, with Twins-ERM and Twins-IRM outperforming all the compared methods.

|  | ERM | IB-ERM | IRM | IB-IRM | Twins-ERM | Twins-IRM | Twins-Direct |
|---|---|---|---|---|---|---|---|
| MNIST | $60.27 \pm 1.21$ | $71.80 \pm 0.69$ | $61.49 \pm 1.45$ | $71.79 \pm 0.70$ | $\mathbf{83.03 \pm 1.34}$ | $82.83 \pm 2.73$ | $79.98 \pm 0.87$ |
| FashionMNIST | $50.92 \pm 1.20$ | $51.74 \pm 1.12$ | $48.41 \pm 0.90$ | $50.92 \pm 1.20$ | $55.60 \pm 3.33$ | $\mathbf{56.04 \pm 1.79}$ | $55.20 \pm 2.16$ |
| KMNIST | $22.80 \pm 1.06$ | $29.21 \pm 0.85$ | $22.89 \pm 0.94$ | $27.83 \pm 0.37$ | $51.24 \pm 3.94$ | $\mathbf{51.52 \pm 3.83}$ | $50.29 \pm 2.58$ |

Table 2: Classification accuracy ($\uparrow$) on MNIST-type datasets. Notice that the proposed Twins-ERM and Twins-IRM exhibit the best performance outperforming previous methods by a significant margin. Twins-Direct (See Appendix C) achieve similar performance with Twins-ERM and Twins-IRM

### 4.3 REAL WORLD DATASETS

In the third benchmark we use DomainBed to experiment on real world datasets: OfficeHome (Venkateswara et al., 2017), PACS (Li et al., 2017). For our Twins algorithm, we ran a hyperparameter search in the range of $\{10^{-4}, 10^{-3}, 10^{-2}, 10^{-1}\}$ for $\mu$, and 20 hyperparameter seeds for remaining hyperparameters in the DomainBed suite (Gulrajani & Lopez-Paz, 2020). We run the experiments using 3 different seeds and report the mean and the standard deviation of the classification. For ERM and IRM, we directly borrow the reuslts from the original paper.

The results are reported in Table 3 and 4. We find that our Twins-IRM algorithm obtain consistent improvement over the baselines.

| Algorithm | A | C | P | R | Avg |
|---|---|---|---|---|---|
| ERM | $61.3 \pm 0.7$ | $52.4 \pm 0.3$ | $75.8 \pm 0.1$ | $76.6 \pm 0.3$ | $66.5 \pm 0.3$ |
| IRM | $58.9 \pm 2.3$ | $52.2 \pm 1.6$ | $72.1 \pm 2.9$ | $74.0 \pm 2.5$ | $64.3 \pm 2.1$ |
| Twins-IRM | $\mathbf{64.8 \pm 0.2}$ | $\mathbf{52.6 \pm 0.7}$ | $\mathbf{77.5 \pm 0.2}$ | $\mathbf{78.9 \pm 0.3}$ | $\mathbf{68.5 \pm 0.4}$ |

Table 3: Classification accuracy ($\uparrow$) on OfficeHome.

| Algorithm | A | C | P | S | Avg |
|---|---|---|---|---|---|
| ERM | $84.7 \pm 0.4$ | $\mathbf{80.8 \pm 0.6}$ | $97.2 \pm 0.3$ | $79.3 \pm 1.0$ | $85.5 \pm 0.7$ |
| IRM | $84.8 \pm 1.3$ | $76.4 \pm 1.1$ | $96.7 \pm 0.6$ | $76.1 \pm 1.0$ | $83.5 \pm 1.1$ |
| Twins-IRM | $\mathbf{88.0 \pm 0.3}$ | $79.6 \pm 0.4$ | $\mathbf{97.9 \pm 0.5}$ | $\mathbf{80.1 \pm 0.9}$ | $\mathbf{86.4 \pm 0.6}$ |

Table 4: Classification accuracy ($\uparrow$) on PACS.

## 5 RELATED WORK

### 5.1 RELATION TO INVARIANT RISK MINIMIZATION

**Relation to IB-ERM/IB-IRM (Ahuja et al., 2021).** Ahuja et al. (2021) introduce the information bottleneck method. However, similar to Tishby et al. (2000), Ahuja et al. (2021) utilize the information

bottleneck to learn a representation that compresses the input as much as possible while preserving all the relevant information about the target label. Ours instead is label-free and tries to preserve the relevant information about the domain-free latent image during compression, and uses another image with the same label but different domain as surrogate for the unknown domain-free latent image. Essentially, setting $\beta = 0$ will reduce our penalty into the one in Ahuja et al. (2021), but $\beta > 1$ can help eliminate the trivial solution that the representation mapping is constant, and is shown to achieve better performance on various datasets.

Previous work has demonstrated that the entropy penalty alone (i.e. $\beta = 0$) might fail in specific case, such as in Section 4 in Ahuja et al. (2021). Nevertheless, our proposed framework does not suffer from this counter-example. We introduce such an example next as a simple classification problem. In each $e \in \mathcal{E}_{\text{train}}$, $Y^e \leftarrow X^e_{\text{inv}} \oplus N^e$ and $X^e_{\text{spu}} \leftarrow Y^e \oplus V^e$, where all the random variables involved are binary valued, noise $N^e, V^e$ are Bernoulli with parameters $q$ (identical across $\mathcal{E}_{\text{train}}$), $c^e$ (varies across $\mathcal{E}_{\text{train}}$) respectively. If $c^e < q$, then in $\mathcal{E}_{\text{train}}$ predictions based on $X^e_{\text{spu}}$ are better than predictions based on $X^e_{\text{inv}}$. If $\Phi$ selects $X^e_{\text{inv}}$, the IB penalty equals $-\lambda H(X^e_{\text{inv}})$; while if $\Phi$ selects $X^e_{\text{spu}}$, the IB penalty equals $H(N^e \oplus V^e) - \lambda H(X^e_{\text{spu}})$. Since $X^e_{\text{spu}} \leftarrow X^e_{\text{inv}} \oplus N^e \oplus V^e$, we have $-\lambda H(X^e_{\text{inv}}) < H(N^e \oplus V^e) - \lambda H(X^e_{\text{spu}})$ by Lemma A.2 in the Appendix. Our IB penalty is then able to select the invariance term $X^e_{\text{inv}}$. On the other hand, if $X^e_{\text{inv}}$ obeys uniform Bernoulli distribution, its entropy will be no lower than $X^e_{\text{spu}}$, and hence the entropy term alone is not enough.

## 5.2 RELATION TO CROSS-DOMAIN COVARIANCE METHOD

Aligning the cross-domain distribution has been studied extensively in the domain generalization community both empirically and theoretically (Sun & Saenko, 2016; Li et al., 2018b; Rahman et al., 2020; Kpotufe & Martinet, 2018). Despite the fact that we use covariance as well, the motivation and implication of the proposed regularization scheme are different. Domain aligning method, such as CORAL (Sun & Saenko, 2016), aligns the correlation matrix from different domains. However, our method (Equation (16)) tries to decorrelate each dimension of the representation by minimizing the off-diagonal term of cross-correlation matrix. In our penalty, we do not want to align the covariance between different domains. For example, given a batch of data of size $B \times N$, where $B$ is the batch size and $N$ is the feature dimension. Cross-domain covariance tries to deal with the row vector and the correlation matrix is of size $N \times N$. Our covariance penalty deals with column vector, and the correlation matrix is of size $B \times B$. Since usually $B \ll N$, the computation would be much easier.

## 5.3 RELATION TO CONTRASTIVE-BASED DOMAIN GENERALIZATION

Our method can also be regarded as a contrastive-based domain generalization problem. Contrastive learning (Chopra et al., 2005; Caron et al., 2020; Grill et al., 2020; Chen et al., 2020; Zbontar et al., 2021) has been a successful paradigm in self-supervised leaning. Contrastive learning aims at bringing positive pair samples closer together, while moving negative samples further away in a learned embedding space. Essentially, the aim of domain generalization is to extract domain-invariant features, similarly aiming to minimize the distance of features within the same class in the embedding space, while maximizing the distance of features from different classes. Such aim is closely related to the domain generalization. SelfReg (Kim et al., 2021) uses only positive data pairs and introduces inter-domain curriculum learning to prevent representation collapse (Grill et al., 2020). (Jeon et al., 2021) uses domain-aware supervised contrastive to ensure domain invariance while increasing class discriminability, Compared to previous works, our method instead introduces a much simpler framework to ensure convergence to domain invariant features with theoretical guarantee.

## 6 CONCLUSION

In this work, we introduce an information-theoretical approach for domain generalization. We cast the task of domain generalization as a rate distortion problem and then use information bottleneck penalty to obtain guarantees on the existence of features we want to learn. We link our method, called Twins, with self-supervised learning, which can provide a theoretical perspective in the success behind self-supervised learning. We conduct an empirical study on Twins-ERM and Twins-IRM under various datasets and confirm the consistent improvement of the proposed method over existing baselines. In the future, we intend to further analyze domain generalization in the rate distortion framework and conduct large scale experiments to verify our IB formulation in real-world applications.

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

## A  PROOF OF THEOREM 3.1

The entropy or the Shannon entropy (Wehrl, 1978) of a discrete random variable $X \sim \mathbb{P}_X$ with support $\mathcal{X}$ is defined as

$$H(X) = -\sum_{x \in \mathcal{X}} \mathbb{P}_X(X = x) \log \big(\mathbb{P}_X(X = x)\big). \tag{19}$$

The differential entropy (Wehrl, 1978) of a continuous random variable $X \sim \mathbb{P}_X$ with support $\mathcal{X}$ is given as follows

$$h(X) = -\int_{x \in \mathcal{X}} \log \big(\mathbb{P}_X(x)\big) d\mathbb{P}_X(x), \tag{20}$$

where $d\mathbb{P}_X(x)$ is the Radon-Nikodym derivative of $\mathbb{P}_X$ w.r.t the Lesbegue measure.

For continuous variables, the differential entropy $h(\cdot)$ is not bounded below, we can define the lower bounded differential entropy (Kirsch et al., 2020) $\widehat{h}(X) = h(X + \varepsilon)$, where $\varepsilon$ is an independent zero-entropy noise $\varepsilon \sim \text{Uniform}(0, 1)$. Since $X \perp \varepsilon, \widehat{h}(X) \geq h(\varepsilon) = 0$, we get that $\widehat{h}(\cdot)$ is bounded below.

**Lemma A.1.** *If $X$ and $Y$ are discrete random variables that are independent with the supports satisfying $2 \leq |\mathcal{X}| < \infty, 2 \leq |\mathcal{Y}| < \infty$, where $|\cdot|$ denotes the number of element in a set, then for $\lambda < 1$,*

$$\lambda H(X) + H(Y) > \lambda H(X + Y) \tag{21}$$

*Proof.* Define $Z = X + Y$.

$$\begin{aligned}
H(Z|X) &= -\sum_{x \in \mathcal{X}} \mathbb{P}_X(x) \sum_{z \in \mathcal{Z}} \mathbb{P}_{Z|X}(Z = z|X = x) \log \Big(\mathbb{P}_{Z|X}(Z = z|X = x)\Big) \\
&= -\sum_{x \in \mathcal{X}} \mathbb{P}_X(x) \sum_{z \in \mathcal{Z}} \mathbb{P}_{Y|X}(Y = z - x|X = x) \log \Big(\mathbb{P}_{Y|X}(Y = z - x|X = x)\Big) \\
&= -\sum_{x \in \mathcal{X}} \mathbb{P}_X(x) \sum_{z \in \mathcal{Z}} \mathbb{P}_{Y|X}(Y = z - x|X = x) \log \Big(\mathbb{P}_{Y|X}(Y = z - x|X = x)\Big) \text{ (use } X \perp Y) \\
&= -\sum_{x \in \mathcal{X}} \mathbb{P}_X(x) \sum_{z \in \mathcal{Z}} \mathbb{P}_Y(Y = z - x) \log \Big(\mathbb{P}_Y(Y = z - x)\Big) \\
&= H(Y)
\end{aligned} \tag{22}$$

Hence,

$$H(X+Y) - H(X) = H(X+Y) - H(X+Y|Y) = I(X+Y; Y) = H(Y) - H(Y|X+Y) \leq H(Y) \tag{23}$$

$$\lambda(H(X + Y) - H(X)) \leq \lambda H(Y) \leq H(Y) \tag{24}$$

when $\lambda < 1$.

The equality holds if and only if $H(Y) = 0$, which is impossible since $2 \leq |\mathcal{Y}| < \infty$. $\qquad \square$

**Lemma A.2.** *If $X$, $Y$ and $Z$ are discrete random variables with the supports satisfying $2 \leq |\mathcal{X}| < \infty, 2 \leq |\mathcal{Y}| < \infty$ and $2 \leq |\mathcal{Y}| < \infty$, where $|\cdot|$ denotes the number of element in a set. Besides, $Y$ is independent of $X$ and $Z$. Then for $\lambda < 1$,*

$$H(X + Y|Z) - \lambda H(X + Y) > H(X|Z) - \lambda H(X) \tag{25}$$

*Proof.* Similar to Equation (22), we would have

$$\begin{aligned}
H(X + Y) - H(X) &= I(X + Y; Y), \\
H(X + Y|Z) - H(X|Z) &= I(X + Y; Y|Z),
\end{aligned} \tag{26}$$

By the chain rule of conditional mutual information,

$$
\begin{aligned}
I(X+Y;Y|Z) &= I(Y;X+Y,Z) - I(Y;Z) \\
&= I(Y;X+Y,Z) - 0 \quad (\text{ since } Y \perp Z) \\
&= I(Y;X+Y) + I(Y;Z|X+Y) \geq I(Y;X+Y)
\end{aligned}
\tag{27}
$$

where the last inequality holds since the conditional mutual information is non-negative. Since $\lambda < 1$,

$$
I(X+Y;Y|Z) \geq I(Y;X+Y) \geq \lambda I(Y;X+Y)
\tag{28}
$$

and hence the equality holds iff $I(Y;X+Y) = 0$, in other words, $Y \perp X+Y$. If given $X+Y = x_{\max} + y_{\max}$, we can infer $Y = y_{\max}$. Hence, $\mathbb{P}(Y = y_{\max}|X+Y = x_{\max} + y_{\max}) = 1$. However, $\mathbb{P}(Y = y_{\max}) = 1$ as the support of $Y$ has at least two elements, which gives a contradiction. Hence,

$$
I(X+Y;Y|Z) > \lambda I(Y;X+Y)
\tag{29}
$$

$\square$

**Lemma A.3.** *If $X$ and $Y$ are continuous random variables that are independent and have a bounded support, then for $\lambda < 1$,*

$$
\widehat{h}(X) + \lambda\widehat{h}(Y) > \lambda\widehat{h}(X+Y)
\tag{30}
$$

*Proof.* Setting $\varepsilon \sim \text{Uniform}(0,1)$ independent of $X, Y$, we have $\widehat{h}(X) = h(X+\varepsilon), \widehat{h}(Y) = h(Y+\varepsilon), \widehat{h}(X+Y) = h(X+Y+\varepsilon)$. We have

$$
\begin{aligned}
\lambda(h(X+Y+\varepsilon) - h(Y+\varepsilon)) &= \lambda I(X+Y+\varepsilon;X) \\
\text{and } h(X+\varepsilon) = h(X+\varepsilon) - h(\varepsilon) &= I(X+\varepsilon;X)
\end{aligned}
\tag{31}
$$

According to the data processing inquality, (Beaudry & Renner, 2011) and $X \perp X+Y+\varepsilon|X+\varepsilon$,

$$
I(X+Y+\varepsilon;X) \leq I(X+\varepsilon,Y;X) = I(X+\varepsilon;X)
\tag{32}
$$

where the last equality holds since $X+\varepsilon \perp Y$, and we get

$$
\lambda(h(X+Y+\varepsilon) - h(Y+\varepsilon)) \leq I(X+Y+\varepsilon;X) \leq I(X+\varepsilon;X) = h(X+\varepsilon)
\tag{33}
$$

Rearranging it, we get

$$
\widehat{h}(X) + \lambda\widehat{h}(Y) \geq \lambda\widehat{h}(X+Y)
\tag{34}
$$

Since $\lambda < 1$, the equality holds only if $I(X+Y+\varepsilon;X) = 0$. In other words, we have $X+Y+\varepsilon \perp X$. In the next, we show that it is not possible.

The support of $X$ can be divided into the union of intervals. We assume $\Delta > 0$ such that $[x_{\max} - \Delta, x_{\max}]$ belongs to the rightmost interval of $X$; and $[y_{\max} - \Delta, y_{\max}]$ belongs to the rightmost interval of $Y$, where $x_{\max}$ and $y_{\max}$ denotes the maximum of the support of $X$ and $Y$. Define an event $\mathcal{M} : x_{\max} + y_{\max} - \delta \leq X+Y+\varepsilon \leq x_{\max} + y_{\max} + 1$. If $\mathcal{M}$ occurs, note that $\varepsilon$ is bounded by 1, we have

$$
\mathbb{P}_X(X \leq x_{\max} - \delta|\mathcal{M}) = 0, \quad \mathbb{P}_Y(Y \leq y_{\max} - \delta|\mathcal{M}) = 0
\tag{35}
$$

If $\delta < \Delta$, based on the definition of the interval, we have that

$$
\mathbb{P}_X(X \leq x_{\max} - \delta) > 0, \quad \mathbb{P}_Y(Y \leq y_{\max} - \delta) > 0
\tag{36}
$$

If $X+Y+\varepsilon \perp Y$ then $\mathbb{P}_Y(Y \leq y_{\max} - \delta) = \mathbb{P}_Y(Y \leq y_{\max} - \delta|\mathcal{M})$, which is not the case from the above equations (35) and (36). $\square$

**Lemma A.4.** *If $X, Y$ and $Z$ are continuous random variables that have a bounded support, and $Y$ is independent of $X$ and $Z$, then for $\lambda < 1$,*

$$
\widehat{h}(X+Y|Z) - \lambda\widehat{h}(X+Y) \geq \widehat{h}(X|Z) - \lambda\widehat{h}(X)
\tag{37}
$$

*Proof.* Like Lemma A.2, we rewrite the inequality as

$$I(X + Y + \varepsilon; Y|Z) > \lambda I(X + Y + \varepsilon; Y) \tag{38}$$

Similar to Equation (27), $Y$ is independent of $Z$ we could write $I(X + Y + \varepsilon; Y|Z) = I(Y; Z, X + Y + \varepsilon)$. We then use the data processing inequality, we would get

$$I(X + Y + \varepsilon; Y|Z) = I(Y; Z, X + Y + \varepsilon) \geq I(X + Y + \varepsilon; Y) \tag{39}$$

since $\lambda < 1$, and similar to the proof of Lemma A.3, $I(X + Y + \varepsilon; Y) > 0$, we would have

$$I(X + Y + \varepsilon; Y|Z) > \lambda I(X + Y + \varepsilon; Y) \tag{40}$$

$\square$

*Proof of Theorem 3.1.* The proof of the theorem resembles the proof of Theorem 4 in (Ahuja et al., 2021). Consider a solution to equation $\Phi^\dagger$,

$$\Phi^\dagger \cdot X^e = \Phi^\dagger \cdot S(X^e_{\text{inv}}, X^e_{\text{spu}}) = \Phi_{\text{inv}} \cdot X^e_{\text{inv}} + \Phi_{\text{spu}} \cdot X^e_{\text{spu}}$$
$$= \Big[\Phi_{\text{inv}} + \Phi_{\text{spu}} \cdot A\Big] \cdot X^e_{\text{inv}} + \Phi_{\text{spu}} \cdot W^e. \tag{41}$$

and since $\Phi^\dagger$ achieves the error of $q$,

$$\text{I}(w^+_{\text{inv}} \cdot X^e_{\text{inv}}) = \text{I}(\Phi_{\text{inv}} \cdot X^e_{\text{inv}} + \Phi_{\text{spu}} \cdot X^e_{\text{spu}}) \tag{42}$$

In the next we prove $\Phi_{\text{spu}}$ by contradiction. Define $\Phi^+ = \Big(\Big[\Phi_{\text{inv}} + \Phi_{\text{spu}} \cdot A\Big], 0\Big) S^{-1}$. Observe that we can write $\Phi^\dagger \cdot X^e = \Phi^+ \cdot X^e + \Phi_{\text{spu}} \cdot W^e$. a) $\Phi_{\text{spu}} \cdot W^e \perp \Phi^+ \cdot X^e$ ($\Phi^+ \cdot X^e = \Big[\Phi_{\text{inv}} + \Phi_{\text{spu}} \cdot A\Big] \cdot X^e_{\text{inv}}$ and $X^e_{\text{inv}} \perp W^e$),

b.1) $\Phi^+ \cdot X^e$, $\Phi_{\text{spu}} \cdot W^e$ are discrete random variables with finite support of size at least two. (discrete case)

b.2) $\Phi^+ \cdot X^e$, $\Phi_{\text{spu}} \cdot W^e$ are continuous bounded random variables. (continuous case)

In the discrete case, from a), b.1), and Lemma A.1 it follows that

$$\lambda H(\Phi^+ \cdot X^e) + H(\Phi_{\text{spu}} \cdot W^e) > \lambda H(\Phi^\dagger \cdot X^e) \tag{43}$$

Rearranging the terms, we have

$$H(\Phi_{\text{spu}} \cdot W^e) - \lambda H(\Phi^\dagger \cdot X^e) > -\lambda H(\Phi^+ \cdot X^e) \tag{44}$$

Since $X^e_{\text{inv}} = G(X_{\text{inv}}, e)$, we have $H(\Phi^\dagger \cdot X^e | X_{\text{inv}}) = H(\Phi_{\text{spu}} \cdot W^e)$ and $H(\Phi^+ \cdot X^e | X_{\text{inv}}) = 0$. Hence, we get

$$H(\Phi^\dagger \cdot X^e | X_{\text{inv}}) - \lambda H(\Phi_{\text{spu}} \cdot W^e) > H(\Phi^+ \cdot X^e | X_{\text{inv}}) - H(\Phi^+ \cdot X^e). \tag{45}$$

and therefore, $\Phi^+ \cdot X^e$ would have a lower penalty. In the continuous case, the argument is similar by invoking a), b.2) and Lemma A.3. $\Phi^+$ can achieve strictly lower penalty than $\Phi^\dagger$.

Following the proof of the first part of Theorem 4 in (Ahuja et al., 2021), we can show that $\Phi^+$ achieves the same error of $q$ in all the training environments. Thus $\Phi^+$ is a strictly better solution $\Phi^\dagger$, which contradicts the optimality of $\Phi^\dagger$. Therefore, it follows that $\Phi_{\text{spu}} = 0$. And hence,

$$\text{I}(w^+_{\text{inv}} \cdot X^e_{\text{inv}}) = \text{I}(\Phi_{\text{inv}} \cdot X^e_{\text{inv}}) \tag{46}$$

Based on Theorem 3 in (Ahuja et al., 2021), if a solution does not rely on spurious features and satisfies equation (46) for all the points in the support, then under the Assumption 3 such a solution solves the domain generalization problem (DG). $\square$

*Proof of Theorem 3.2.* The major difference here is that we do not condition on the unknown invariant feature. Notations are defined similarly as in the proof of Theorem 3.1. Compared with Equation (45), we now want to prove that:

$$H(\Phi^\dagger \cdot X^e | X^{e'}) - \lambda H(\Phi_{\text{spu}} \cdot W^e) > H(\Phi^+ \cdot X^e | X^{e'}) - H(\Phi^+ \cdot X^e). \tag{47}$$

In other words,

$$H(\Phi^+ \cdot X^e + \Phi_{\text{spu}} \cdot W^e | X^{e'}) - \lambda H(\Phi_{\text{spu}} \cdot W^e) > H(\Phi^+ \cdot X^e | X^{e'}) - H(\Phi^+ \cdot X^e). \tag{48}$$

Since $\Phi_{\text{spu}} \cdot W^e$ is independent of $X^{e'}$ and $\Phi^+ \cdot X^e$. We use Lemma A.2 for the discrete case and Lemma A.4 for the continuous case to prove Equation (48). The rest of the proof is the same as the proof in Theorem 3.1. $\qquad\square$

# B    DATASET DESCRIPTION

## B.1    LINEAR UNIT TEST

**Example 1/1s** The dataset in environment $e \in \mathcal{E}_{\text{all}}$ is sampled from the following distributions:

$$
\begin{aligned}
X_{\text{inv}}^e &\sim \mathcal{N}_{d_{\text{inv}}}(0, (\sigma^e)^2), &\quad \tilde{Y}^e &\sim \mathcal{N}_{d_{\text{inv}}}(W_{yx} X_{\text{inv}}^e, (\sigma^e)^2), \\
X_{\text{spu}}^e &\sim \mathcal{N}_{d_{\text{spu}}}(W_{xy} \tilde{Y}^e, 1), &\quad X^e &\leftarrow S \cdot (X_{\text{inv}}^e, X_{\text{spu}}^e), \\
Y^e &\leftarrow \frac{2}{(d_{\text{inv}} + d_{\text{spu}})} \mathbf{1}_{d_{\text{inv}}}^{\text{T}} \tilde{Y}^e,
\end{aligned}
\tag{49}
$$

where $W_{yz} \in \mathbb{R}^{d_{\text{inv}} \times d_{\text{inv}}}$, $W_{xy} \in \mathbb{R}^{d_{\text{spu}} \times d_{\text{inv}}}$ are matrices drawn i.i.d. from the standard normal distribution, $\mathbf{1}_{d_{\text{inv}}} \in \mathbb{R}^{d_{\text{inv}}}$ is a vector of ones, $\mathcal{N}_k$ is a $k$ dimensional vector from the normal distribution, and $S \in \mathbb{R}^{(d_{\text{inv}} + d_{\text{spu}}) \times (d_{\text{inv}} + d_{\text{spu}})}$ is a rotation matrix fixed for all environments. The parameter $\sigma$ is set differently for every environment (i.e., domain). In particular, we set $(\sigma^{e=e_0})^2 = 0.1$, $(\sigma^{e=e_1})^2 = 1.5$, and $(\sigma^{e=e_2})^2 = 2$ for the first three environments. In case there are more than three environments, the $(\sigma^{e=e_j})$ for $j > 3$ is uniformly from $\text{Unif}(10^{-2}, 10)$. The rotation matrix $S$ is set to the identity matrix in Example 1 and a random unitary matrix in Example 1s.

**Example 2/2s** Following the notation of the original paper (Aubin et al., 2021), let

$$
\begin{aligned}
\mu_{\text{cow}} &= \mathbf{1}_{d_{\text{inv}}}, &\quad \mu_{\text{camel}} &= -\mu_{\text{cow}}, &\quad \nu_{\text{animal}} &= 10^{-2}, \\
\mu_{\text{grass}} &= \mathbf{1}_{d_{\text{spu}}}, &\quad \mu_{\text{sand}} &= -\mu_{\text{grass}}, &\quad \nu_{\text{background}} &= 1.
\end{aligned}
\tag{50}
$$

The dataset in environment $e \in \mathcal{E}_{\text{all}}$ is sampled from the following distribution:

$$
\begin{aligned}
j^e &\sim \text{Categorical}\big(p^e s^e, (1-p^e)s^e, p^e(1-s^e), (1-p^e)(1-s^e)\big), \\
X_{\text{inv}}^e &\sim \begin{cases} (\mathcal{N}_{d_{\text{inv}}}(0, 0.1) + \mu_{\text{cow}}) \cdot \nu_{\text{animal}} & \text{if } j^e \in \{1, 2\}, \\ (\mathcal{N}_{d_{\text{inv}}}(0, 0.1) + \mu_{\text{camel}}) \cdot \nu_{\text{animal}} & \text{if } j^e \in \{3, 4\}, \end{cases} \\
X_{\text{spu}}^e &\sim \begin{cases} (\mathcal{N}_{d_{\text{spu}}}(0, 0.1) + \mu_{\text{grass}}) \cdot \nu_{\text{background}} & \text{if } j^e \in \{1, 4\}, \\ (\mathcal{N}_{d_{\text{spu}}}(0, 0.1) + \mu_{\text{sand}}) \cdot \nu_{\text{background}} & \text{if } j^e \in \{2, 3\}, \end{cases} \\
X^e &\leftarrow S \cdot (X_{\text{inv}}^e, X_{\text{spu}}^e), \quad Y^e \leftarrow \text{I}(\mathbf{1}_{d_{\text{inv}}}^{\text{T}} X_{\text{inv}}^e),
\end{aligned}
\tag{51}
$$

where the environment foreground/background probabilities are $p^{e=e_0} = 0.95$, $p^{e=e_1} = 0.97$, $p^{e=e_2} = 0.99$ and the cow/camel probabilities are $s^{e=e_0} = 0.3$, $s^{e=e_1} = 0.5$, $s^{e=e_2} = 0.7$. For $n_{\text{env}} > 3$ and $j \in [3 : n_{\text{env}} - 1]$, the extra environment variables are respectively drawn according to $p^{e=e_j} \sim \text{Unif}(0.9, 1)$ and $s^{e=e_j} \sim \text{Unif}(0.3, 0.7)$. The rotation matrix $S$ is set to the identity matrix in Example 2 and a random unitary matrix in Example 2s.

**Example 3/3s** The example is meant to present a linear version of the spiral classification problem of Parascandolo et al. (2020). Let $\mu_{\text{inv}} = 0.1 \cdot \mathbf{1}_{d_{\text{inv}}}$, and $\mu_{\text{spu}}^e \sim \mathcal{N}_{d_{\text{spu}}}(0, 1)$ for all the environments. The dataset in environment $e \in \mathcal{E}_{\text{all}}$ is sampled from the following distribution:

$$
\begin{aligned}
Y^e &\sim \text{Bernoulli}\Big(\frac{1}{2}\Big), &\quad X^e &\leftarrow S \cdot (X_{\text{inv}}^e, X_{\text{spu}}^e) \\
X_{\text{inv}}^e &\sim \begin{cases} \mathcal{N}_{d_{\text{inv}}}(+\mu_{\text{inv}}, 0.1) \text{ if } Y^e = 0, \\ \mathcal{N}_{d_{\text{inv}}}(-\mu_{\text{inv}}, 0.1) \text{ if } Y^e = 1, \end{cases} &\quad X_{\text{spu}}^e &\sim \begin{cases} \mathcal{N}_{d_{\text{spu}}}(+\mu_{\text{spu}}^e, 0.1) \text{ if } Y^e = 0, \\ \mathcal{N}_{d_{\text{spu}}}(-\mu_{\text{spu}}^e, 0.1) \text{ if } Y^e = 1, \end{cases}
\end{aligned}
\tag{52}
$$

The rotation matrix $S$ is set to the identity matrix in Example 3 and a random unitary matrix in Example 3s. In the above dataset, the invariant features are anti-causally related to the label $Y^e$.

**Remark on Linear unit test**: In the Example 1/1s and Example 3/3s, the invariant features are causal and partially informative about the label. The spurious features carry extra information about the label not contained in the invariant features. In the Example 2/2s, the invariant features are causal and carry full information about the label.

## C    GAUSSIAN-FREE ENTROPY ESTIMATION

### C.1    ESTIMATING ENTROPY BY kNN

Since the feature is in high-dimensional spaces it is challenging to estimate the density of $Z$, preventing us from directly computing the exact entropy. To remedy this issue, we resort to the particle-based entropy estimator from Singh et al. (2003); Beirlant et al. (1997), which is based on $k$-Nearest Neighbors ($k$NN). We introduce this approach in general terminologies. Consider a distribution $p$ with respect to $z \in \mathcal{Z}$, the particle based entropy estimate is given by

$$\widehat{H}_k(p) = -\frac{1}{N} \sum_{i=1}^{N} \log \frac{k}{N \text{Vol}_i^k} + \log k - \Phi(k) \propto \sum_{i=1}^{N} \log \text{Vol}_i^k \tag{53}$$

where $\Phi$ is the digamma function, $\log k - \Phi(k)$ is a bias correction term. $\text{Vol}_i^k$ is the volume of the hyper-sphere of radius $R_i = \|z_i - z_k^{\text{KNN}}\|_2$, which is the Euclidean distance between $z_i$ and its $k$-th nearest neighbor $z_k^{\text{KNN}}$. The volumn is given by:

$$\text{Vol}_i^k = \frac{\|z_i - z_k^{\text{KNN}}\|_2^n \cdot \pi^{n/2}}{\Gamma(\frac{n}{2} + 1)} \tag{54}$$

where $\Gamma$ is the Gamma function, $n$ is the dimension of $\mathcal{Z}$. Putting Equation (53) and Equation (54) together, we have

$$\widehat{H}_k(p) = \frac{n}{N} \sum_{i=1}^{N} \log \|z_i - z_k^{\text{KNN}}\|_2 + \log N + C \tag{55}$$

where $C_{k,n}$ is determined by $n$ and $k$.

### C.2    ESTIMATING IB OBJECTIVE 4

Recall that $Z^e = f(X^e)$, and $X^e = G(X, e)$, we can estimate the IB objective $L_{IB}$ by first sampling $\{e_i\} \sim \mathbb{P}_e, 1 \leq i \leq D$, and then $X_j^{e_i} \sim \mathbb{P}^{e_i}(X^{e_i}), 1 \leq j \leq N_i$ for any fixed $i$. We can use samples $Z_j^{e_i} = f(X_j^{e_i})$ to estimate $\mathbb{E}_{X,e} H(Z^e)$, i.e.

$$\mathbb{E}_{X,e} H(Z^e) \approx \frac{n}{\sum_{i=1}^{D} N_i} \sum_{i=1}^{D} \sum_{j=1}^{N_i} \log \|Z_j^{e^i} - Z_k^{\text{KNN}}\|_2 + \log(\sum_{i=1}^{D} N_i) + C_{k,n} \tag{56}$$

where $n$ is the dimension of $Z_i$, $Z_k^{\text{KNN}}$ is $Z_j^{e^i}$'s $k$-th nearest neighbor in the full dataset $\{Z_j^{e_i}\}_{i=1,j=1}^{D,N_i}$.

For the first term, we use the same method but conditioned on the fixed label. In other words, it should be

$$\mathbb{E}_{X,e} H(Z^e|X) \approx \frac{1}{D} \sum_{i=1}^{D} \left( \frac{n}{N_i} \sum_{j}^{N_i} \log \|Z_{ij} - Z_{ik}^{\text{KNN}}\|_2 + \log N_i \right) + C_{k,n} \tag{57}$$

where $Z_{ik}^{\text{KNN}}$ is $Z_{ij}$'s $k$-th nearest neighbor in the dataset $\{Z_{ij}\}_j$.

### C.3    COMPARISON

We perform our experiments on ColoredMNIST datasets. At each checkpoint, we sample 1024 isntances and set kNN parameter to be 5 to estimate the IB penalty (4) by Equation (56) and (57) at every checkpoints. We plot the trajectory of kNN based penalty in Figure (1). Clearly, our Twins method is able to efficiently minimize the true Gaussian entropy.

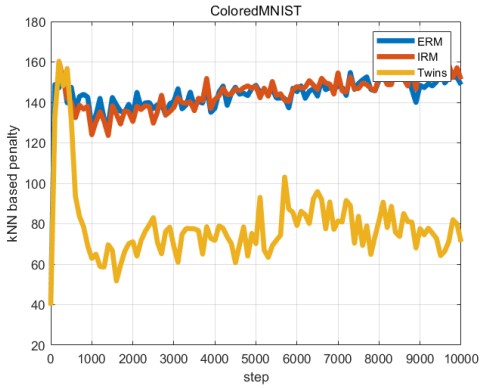

Figure 1: kNN based IB penalty (4)

## D  HYPERPARAMETER SELECTION OVER $\mu$

|  | $\mu = 10^{-4}$ | $10^{-3}$ | $10^{-2}$ | $10^{-1}$ | 1 |
|---|---|---|---|---|---|
| MNIST | $79.24 \pm 0.69$ | $80.44 \pm 2.70$ | $82.83 \pm 2.73$ | $77.19 \pm 6.49$ | $72.11 \pm 4.08$ |
| KMNIST | $49.48 \pm 4.27$ | $52.24 \pm 3.94$ | $52.29 \pm 3.26$ | $43.02 \pm 0.68$ | $36.31 \pm 3.20$ |
| FashionMNIST | $54.88 \pm 1.57$ | $53.87 \pm 2.41$ | $56.04 \pm 1.79$ | $52.25 \pm 2.70$ | $50.96 \pm 1.61$ |

