# OpenReview forum: "A Rate-Distortion Approach to Domain Generalization"
_ICLR.cc/2022/Conference — ICLR 2022 Submitted_

### Official Review · Reviewer_t26w · 2021-11-02

**Correctness:** 2
**Technical Novelty And Significance:** 2
**Empirical Novelty And Significance:** 3
**Recommendation:** 3
**Confidence:** 5

**Main Review:**

**Strengths:** The problem addressed in the paper is very important. It is a good idea to combine the ideas introduced in model based domain generalization [Robey et al.] and the ideas of information bottleneck and invariance [Ahuja et al.].  The experiments show some promise. The paper is easy to read as well.


**Weaknesses:** I highlight concerns with the paper below in a pointwise fashion

1. **Access to X to prove theorems makes theorems vacuous** The authors theory assumes access to domain invariant label. If we already have access to X to prove the theorems, the theorems are vaccuous because you can just use X to learn the predictor in any domain and that should work fine. I will describe a data generation process to further my point.

   $X^e \leftarrow G(X,e)$

   $Y^e \leftarrow f(X) + N $

  In the above model the whole difficulty arises because we have access to $X^e$ and not $X$. If we have access to $X$ and have $f$ in the space of functions that the learner searches, then with enough data $f$ would be learned given $X$ as input. Therefore, the theorem in the current version do not make much sense.

2. **Proxy for X is not reliable** The authors for the experiments do not assume $X$ is available and instead rely on a proxy. The proxy is that for an instance $X^e$  the authors use an alternate $X^{e'}$ from a different environment but with the same label. If the authors are borrowing the idea from Robey et al. already why do they not use the $G$ function from [Robey et al.] and use the domain invariant $X$. Take the $G$ model from Robey et al. for each $X^{e}$ find the alternate $X^{e'}$ using their model. Table 1 in [Robey et al.] very clearly illustrates what I am saying. Using the learned model $G$ seems a much more appropriate proxy and not just any point from another domain with the same label.

3. **Regarding experiments** I would encourage the authors to modify their method to incorporate $G$ model in it. However, if the authors insist that their approach is already working and robust, then some more experiments and justification (ablation studies for the same are needed).  Take datasets such as Terra Incognita, Camelyon etc. used in DomainBed and WILDS repository and show that the idea continues to work in more diverse settings.



**Summary Of The Paper:**

In this work, the authors study the problem of domain generalization. Recent works such as invariant risk minimization [Arjovsky et al.] have invigorated a lot of interest in the problem of domain generalization. The paper brings together ideas from rate distortion and information bottleneck principle (recently used in Ahuja et al. for domain generalization) and the work of model based domain generalization [Robey et al.].
In Ahuja et al., the authors combine invariance and information bottleneck to address domain generalization. The representation learned in Ahuja et al. is constrained by the label and information bottleneck constraints. In this work, the authors add some more structure to the generation of the observations X from the latent and say that there is a common instance X that is transformed by the domain transformation model to get the image in the current domain, i.e. X^e = G(X,e). The authors try to recover the common instance X that is shared across domains.



**Summary Of The Review:**

Overall, the paper proposes an interesting approach to combine information bottleneck and model based domain generalization ideas recently used in [Ahuja et al.] and [Robey et al.]. However, there are severe shortcomings in the paper, the theory seems vacuous under assumption of access to $X$ (authors should please correct this). The proxy for $X$ is not reliable and instead $G$ model from Robey et al. should be used to get good results in large scale settings.

---

> ### Author Response · Authors · 2021-11-22
> **Reply to reviewer t26w**
>
>  We thank the reviewer for the feedback. We address their concerns below:
>
> - **Access to X to prove theorems makes theorems vacuous**  We are thankful to the reviewer for pointing this out. We have proved a new theorem that does not rely on the domain-free latent X. Instead, the proxy from another domain $X^{e'}$ would suffice. Such theorem would ensure the validity of using instances from another domain to construct contrastive batch. Please see Theorem 3.2.
>
> - **Proxy for X is not reliable** & **Regarding experiments** We performed contrastive batch construction method by explicitly learning the domain transformation model $G$ under the MUNIT framework. Due to the computational resources, we only performed experiments on the PACS datasets.  We ran a hyperparameter search in the range of $\{10^{-4},10^{-3}, 10^{-2},10^{-1}\}$ for $\mu$, and 5 hyperparameter seeds for other hyperparameters in the DomainBed suite.  We find that using $G$ models are not as good as tbe shuffling. G model's best avg accuracy is ~0.8, while shuffling's best avg accuracy is ~0.86. We will add more numerical results of the $G$ model in the next week.

---

> > ### Author Response · Authors · 2021-11-29
> > **Any questions from the reviewer t26w?**
> >
> > We are thankful to the reviewer for pointing out the weakness in our proof (i.e. assuming access to $X$). This has led to a proving a new theorem that does not rely on $X$, which has improved our work. Given that this was one of the core concerns, we are wondering whether the opinion of the reviewer about our work has improved.

---

> > > ### Author Response · Authors · 2021-11-29
> > > **Additional Reply to reviewer t26w**
> > >
> > > In order to better compare the results of different methods, we use MNIST-type datasets to perform validation  of the explicit domain transformation model G, denoted as TWINS-G. We observe no performance improvement in these three datasets either. Hence, we conclude that it is not necessary to introduce the domain transformation model explicitly and a simple shuffling would suffice.
> > >
> > >
> > > |              | ERM              | IB-ERM           | IRM              | IB-IRM           | Twins-ERM        | Twins-IRM        | Twins-Direct    | TWINS-G          |
> > > | ------------ | ---------------- | ---------------- | ---------------- | ---------------- | ---------------- | ---------------- | --------------- | ---------------- |
> > > | MNIST        | 60.27 $\pm$ 1.21 | 71.80 $\pm$ 0.69 | 61.49 $\pm$ 1.45 | 71.79 $\pm$ 0.70 | **83.03 $\pm$ 1.34** | 82.83 $\pm$ 2.73 | 79.98$\pm$ 0.87 | 74.39 $\pm$6.60  |
> > > | FashionMNIST | 50.92 $\pm$ 1.20 | 51.74 $\pm$ 1.12 | 48.41 $\pm$ 0.90 | 50.92 $\pm$ 1.20 | 55.60 $\pm$ 3.33 | **56.04 $\pm$ 1.79** | 55.20$\pm$ 2.16 | 50.29 $\pm$ 3.47 |
> > > | KMNIST       | 22.80 $\pm$ 1.06 | 29.21 $\pm$ 0.85 | 22.89 $\pm$ 0.94 | 27.83 $\pm$ 0.37 | 51.24 $\pm$ 3.94 | **51.52 $\pm$ 3.83** | 50.29$\pm$ 2.58 | 50.86 $\pm$ 4.89 |

---

### Official Review · Reviewer_38sD · 2021-11-03

**Correctness:** 3
**Technical Novelty And Significance:** 3
**Empirical Novelty And Significance:** 2
**Recommendation:** 6
**Confidence:** 5

**Main Review:**

Weakness:

1) Authors strongly suggest that learning domain invariant features is the only way to solve the domain generalization problem. They prove that  information bottleneck penalty guarantees that domain-invariant features can be learned which in turn guarantees that there is a solution to the domain generalization problem. This may not be completely correct as authors missed out on some of the literature.

2) Weak literature survey. Please cite the papers [1-6] that may be missing from the current paper.

3) Weak experiments section.

Strength:

1) Clear assumption and claims (even through they are strong in some ways, clarity in writing helps understand what authors are trying to convey).

2) Strong justification on why their method could work.

3) Experiments with synthetic data for better understanding.

Main Review:

1) “Domain generalization deals with the difference in the distribution between the training and testing datasets, i.e., the domain shift problem, by extracting domain-invariant features”: This is not correct. Extracting domain-invariant features is just one way to address the domain generalization problem.

2) “The literature on domain generalization aims to learn the features that are invariant across multiple training domains, and to use those features for out-of-distribution generalization.”: Again not correct. Please look at section 2 of Li et al 2019 [1] to get an idea of some non domain invariant learning methods. You could also take a look at [2-5]. These references also should be cited in the current paper.

3) Assumption 1 and invariant feature map: In “Domain Generalization by Marginal Transfer Learning”, there is no assumption like assumption 1 in the paper. Does this assumption 1 help in anyway to tighten the learning theoretic bound in terms of number of source domains and number of training examples available in the source domains?

4) Assumption 4: “Strictly separable invariant features” This looks like a very strong assumption. What happens if this assumption is not satisfied all the time for all the source domains?

5) The experimental section is not complete as it does not contain most datasets that recent papers use: For example, PACS or VLCS datasets are missing [6]. Can authors get results on one of these datasets at least?



[1] Li, Yiying, Yongxin Yang, Wei Zhou, and Timothy Hospedales. "Feature-critic networks for heterogeneous domain generalization." In International Conference on Machine Learning, pp. 3915-3924. PMLR, 2019.

[2] Deshmukh, Aniket Anand, Ankit Bansal, and Akash Rastogi. "Domain2vec: Deep domain generalization." arXiv preprint arXiv:1807.02919 (2018).

[3] Niu, Li, Wen Li, and Dong Xu. "Multi-view domain generalization for visual recognition." In Proceedings of the IEEE international conference on computer vision, pp. 4193-4201. 2015.

[4] Blanchard, Gilles, Aniket Anand Deshmukh, Ürün Dogan, Gyemin Lee, and Clayton Scott. "Domain Generalization by Marginal Transfer Learning." arXiv preprint arXiv:1711.07910 (2017) J. Mach. Learn. Res. (JMLR) 22 (2021): 2-1.

[5] Duan, Lixin, Ivor W. Tsang, and Dong Xu. "Domain transfer multiple kernel learning." IEEE Transactions on Pattern Analysis and Machine Intelligence 34, no. 3 (2012): 465-479.

[6] Li, Da, Yongxin Yang, Yi-Zhe Song, and Timothy M. Hospedales. "Deeper, broader and artier domain generalization." In Proceedings of the IEEE international conference on computer vision, pp. 5542-5550. 2017.


**Summary Of The Paper:**

Authors address a problem of domain generalization using  information bottleneck method. Authors assume that learning domain invariant features is a key to solve the domain generalization problem and then prove that  information bottleneck penalty guarantees that domain-invariant features can be learned.

**Summary Of The Review:**

Authors should address the weaknesses of the paper. Improved literature survey and comparing/contrasting with suggested papers will help improve the score. Stronger experiments section can further improve the score.

---

> ### Author Response · Authors · 2021-11-22
> **Reply to reviewer 38sD**
>
> We thank the reviewer for the feedback. We address their concerns below:
>
> - **Non Domain-Invariant Learning**  Fixed. We have replaced the sentence:"Domain generalization deals with the difference in the distribution between the training and testing datasets, i.e., the domain shift problem, by extracting domain-invariant features" with the following sentence: "Learning features that are invariant across multiple training domains, and using those features for out-of-distribution generalization has emerged as a significant topic in domain generalization" to include non domain-invariant learning method. We also add a paragraph in Section 1 to include more references about non domain invariant learning methods.
>
> - **Assumption 1** We add description about why we introduce the Assumption 1 in Section 1. The Assumption1 can somewhat reflect the generation of domain specific instances. For example, the multiple different views of a 3D object [1], different rotation angles of the image (Rotated MNIST [2]). Besides, the MUNIT architecture [3]  can effectively distangle the domain-free latent instance $X$ and the specific environment $e$, and thus can be used as the domain transformation model $G$ [4].
>
> - **Strictly separable invariant features** We added a mathematical description of such assumption: $\min\_{x\in \bigcup\_{e\in \mathrm{Etrain}} \mathcal{X}\_{\rm inv}^e} \mathrm{sign}(w\_{\mathrm{inv}}^{\star}\cdot x)\cdot (w\_{\mathrm{inv}}^{\star}\cdot x)>0$. Such assumption also appears in [5]. From our perspective, in the data generating process: $I(w_{\rm inv}^\star \cdot X_{\rm inv})$, we would not expect $w_{\rm inv}^\star\cdot X_{\rm inv}$ to be zero (i.e. on the borderline) for any invariant feature $X_{\rm inv}\in\bigcup_{e\in \mathrm{Etrain}} \mathcal{X}\_{\rm inv}^e$. Note that since noise $N^e$ is added, separation of invariant feature is not relevant to the label $y^e$, i.e. the datasets $(X^e, y^e)$ **cannot** be separated by the plane $w_{\rm inv}^*$, such that $y^e$ is identical on one side.
>
> - **Real-world Datasets** We have conducted experiments on PACS and OfficeHome in Section 4.3. Please see the replies to Reviewer eG9q for the numerical tables.
>
> [1] Li  Niu,  Wen  Li,  and  Dong  Xu.   Multi-view  domain  generalization  for  visual  recognition
>
> [2] Daniel E Worrall, Stephan J Garbin, Daniyar Turmukhambetov, and Gabriel J Brostow. Harmonicnetworks: Deep translation and rotation equivariance
>
> [3] Xun Huang, Ming-Yu Liu, Serge Belongie, and Jan Kautz. Multimodal unsupervised image-to-imagetranslation.
>
> [4] Alexander Robey, George J Pappas, and Hamed Hassani. Model-based domain generalization.
>
> [5] Kartik Ahuja, Ethan Caballero, Dinghuai Zhang, Yoshua Bengio, Ioannis Mitliagkas, and Irina Rish.Invariance principle meets information bottleneck for out-of-distribution generalization.

---

> > ### Comment · Reviewer_38sD · 2021-11-29
> > **Score updated**
> >
> > There are still more papers that authors should cite that authors missed from previous list. Note that list suggested is not exhaustive and authors should look for seminal papers in the domain generalization area [1-3].
> >
> >
> > Authors have addressed some of the issues (Non Domain-Invariant Learning) and added experiments on PACS dataset.  I am improving my score to 6.
> >
> > [1] Blanchard, Gilles, Aniket Anand Deshmukh, Ürün Dogan, Gyemin Lee, and Clayton Scott. "Domain Generalization by Marginal Transfer Learning." arXiv preprint arXiv:1711.07910 (2017) J. Mach. Learn. Res. (JMLR) 22 (2021): 2-1.
> >
> > [2] Erfani, Sarah, Mahsa Baktashmotlagh, Masud Moshtaghi, Xuan Nguyen, Christopher Leckie, James Bailey, and Rao Kotagiri. "Robust domain generalisation by enforcing distribution invariance." In Proceedings of the Twenty-Fifth International Joint Conference on Artificial Intelligence (IJCAI-16), pp. 1455-1461. AAAI Press, 2016.
> >
> > [3] Grubinger, Thomas, Adriana Birlutiu, Holger Schöner, Thomas Natschläger, and Tom Heskes. "Domain generalization based on transfer component analysis." In International Work-Conference on Artificial Neural Networks, pp. 325-334. Springer, Cham, 2015.

---

> > > ### Author Response · Authors · 2021-11-29
> > > **Response to the reviewer 38sD**
> > >
> > > We are thankful to the reviewer for appreciating our effort and the revision. We will add the related work in the next update of our work.

---

### Official Review · Reviewer_DWwT · 2021-11-03

**Correctness:** 3
**Technical Novelty And Significance:** 2
**Empirical Novelty And Significance:** 2
**Recommendation:** 5
**Confidence:** 3

**Main Review:**

- It is good that the paper provides the theoretical gurantee of the algorithm for linear binary classiciation.

- After several approximations, the actual contrastive loss formulation in Eq.(14) is adopted. This formulation is closely related to some existing works [A,B] that also employ the constrastive loss for domain generalization. Since they share the similar idea, proper discussion and comparison with them will be necessary.

- The batch construction method and the batch size will directly affect the loss Eq.(14) used for the training. For example, the amount of hard examples included in the contrastive pairs will depend on these choices. It would be good to include the experiemtnal analysis as well as discussion on these factors.

- It is good that the proposed method consistently improves the corresponding baseline methods in two experiments, linear unit tests and mnist-type datasets. However, I have concern that evaluation on more realistic datasets is missing. As the method assumes Gaussian bottleneck which could be a over-simplification for real world problems, I think evaluation on other datasets, such as VLCS, PACS, and OfficeHome, in domainbed would be helpful to show the generalization performance of the method.

- How does the hyperparemter mu affects the accruacy? Analysis of the effect of hyperparameter is missing.

[A] Feature Stylization and Domain-aware Contrastive Learning for Domain Generalization, ACM MM 2021.
[B] SelfReg: Self-supervised Contrastive Regularization for Domain Generalization, ICCV 2021.

**Summary Of The Paper:**

The paper proposes a method for domain generalization. They formulate the problem as a rate-distortion problem and introduce the information bottleneck penalty. Specifically, they realize the objective as a contrastive loss that encourges embeddings across domains to be similar if they share the same label. In the experiments, the method improves the corresponding baselines in several toy datasets.

**Summary Of The Review:**

The paper proposes an interesting approach based on the information bottleneck. However, I think the discussion and comparison with related work on more realistic datasets is necessary.

---
After authors' response
I appreciate the authors' feedback. They provided additional results on more realistic datasets (PACS and OfficeHome) and results for different $\mu$. After reading the feedback as well as other reviews, I would like to keep my score. As also mentioned by other reviewers, the theoretical guarantee of the algorithm is based on quite strong assumptions, which do not likely hold under practical conditions, and I am a bit skeptical about its significance.

---

> ### Author Response · Authors · 2021-11-22
> **Reply to reviewer DWwT**
>
> We thank the reviewer for the feedback. We address their concerns below:
>
> - **Relations to Prior Work** Thanks for pointing out. We add discussions and comparisons in Section 5.3 between our method and prior ones. [1, 2]
>
> - **Batch construction method** In our paper, we construct the contrastive batch by shuffling the batch of instances. As being suggested by Reviewer t26w, we introduce another batch construction method by explicitly learning the domain transformation model $G$ under the MUNIT framework [3]. However, such $G$ model does not help. We ran a hyperparameter search in the range of $\{10^{-4},10^{-3}, 10^{-2},10^{-1}\}$ for $\mu$, and 5 hyperparameter seeds for other hyperparameters in the DomainBed suite.  We find that using $G$ models are not as good as tbe shuffling. G model's best avg accuracy is ~0.8, while shuffling's best avg accuracy is ~0.86. The selection  of batch size has been included in the DomainBed framework. To be more specific, the batch size is sampled from $\mathrm{int}(2^{{\rm Uniform}(3, 5.5)})$, with default value 32.
>
> - **Real-world Datasets** We have conducted experiments on PACS and OfficeHome in Section 4.3. Please see the replies to Reviewer eG9q.
>
> - **Analysis of $\mu$** We have performed hyperparameter selection over $\mu$ within $\{10^{-4}, 10^{-3}, 10^{-3}, 10^{-1}, 1\}$. Generally, we find that the optimal $\mu$ is around $10^{-2}$.
>
>   |              | $\mu=10^{-4}$ | $10^{-3}$     | $10^{-2}$         | $10^{-1}$     | 1             |
>   | ------------ | ------------- | ------------- | ----------------- | ------------- | ------------- |
>   | MNIST        | 79.24 ± 0.69  | 80.44 +- 2.70 | **82.83 ± 2.73**  | 77.19 +- 6.49 | 72.11 +- 4.08 |
>   | KMNIST       | 49.48+- 4.27  | 52.24 +- 3.94 | **52.29 +- 3.26** | 43.02 +- 0.68 | 36.31 +- 3.20 |
>   | FashionMNIST | 54.88 +- 1.57 | 53.87 +- 2.41 | **56.04 ± 1.79**  | 52.25 +- 2.70 | 50.96 +- 1.61 |
>
> [1] Feature Stylization and Domain-aware Contrastive Learning for Domain Generalization, ACM MM 2021.
>
> [2] SelfReg: Self-supervised Contrastive Regularization for Domain Generalization, ICCV 2021.
>
> [3] Multimodal Unsupervised Image-to-Image Translation, ECCV 2018.

---

> > ### Author Response · Authors · 2021-11-29
> > **Any comments from the reviewer DWwT?**
> >
> > We are thankful to the reviewer for their constructive feedback. We have added a number of experiments during the response period to address the concern on more realistic datasets (i.e. PACS and OfficeHome), and we plan to include more experiments in the updated version of the paper. The aforementioned experiments verify that Twins-IRM outperforms the baselines. Given the upcoming discussion-ending deadline, we are wondering whether the concerns of the reviewer have been addressed or there are any remaining questions we can answer.

---

### Official Review · Reviewer_eG9q · 2021-11-05

**Correctness:** 3
**Technical Novelty And Significance:** 2
**Empirical Novelty And Significance:** 1
**Recommendation:** 5
**Confidence:** 4

**Details Of Ethics Concerns:**

N/A.

**Main Review:**

Strengths:
- The paper is well-written and presents the central contributions clearly.
- The authors list the assumptions in the data-generating process and the algorithm (Section 3) quite clearly.
- The approach is developed following a consistent and a (mostly) rigorous line of argumentation, with some theoretical guarantees on convergence under suitable assumptions.

Weaknesses:
- In my opinion, while this work presents some good arguments, its central contributions are not that distinctive from prior work. Minimizing some notion of cross-domain covariance has been studied extensively in both the machine learning and computer vision communities, to varying levels of success (see, e.g., https://openaccess.thecvf.com/content_cvpr_2018/papers/Li_Domain_Generalization_With_CVPR_2018_paper.pdf, https://www.sciencedirect.com/science/article/abs/pii/S003132031930425X for practical examples, and the papers from Samory Kpotufe's group, e.g., https://arxiv.org/abs/1803.01833 that study this in the context of domain adaptation). The central idea is not really inspiring, albeit the connections to rate distortion theory are interesting. Furthermore, it has been shown recently that for real-world problems of domain generalization, it is very hard to beat naive ERM [https://arxiv.org/abs/2007.01434] which makes me wonder whether the assumptions in this paper (and more generally, in a lot of invariant feature research) really hold in practice.
- This brings me to my second point: The contributions within this paper (both theoretical and the experimental results) do not really further any insight into the domain generalization problem IMHO. The assumptions the authors make are far too simplistic for real-world settings (e.g., zero-main Gaussian feature distributions, which are patently not true for image classification that regularly exhibits long-tailed label distributions), and hence do not provide a complete picture, and can also be at odds with prior work. The experimental results are on handcrafted problems and toy datasets, which again, are too simplistic (and in the case of handcrafted problems, not generalizable to real-world problems). The problem of domain generalization suffers specifically from this issue, where a lot of the proposed work is valid only under certain restrictive assumptions, and usually fail when deployed in practice. It would be nice to see if these assumptions were validated by experiments on data that is not carefully curated and drawn from real-world problems.

**Summary Of The Paper:**

This paper studies the problem of domain generalization via invariant feature learning studied from the lens of information theory. The motivation is to use rate distortion theory to obtain a domain-invariant representation. The authors propose an algorithm titled Twins that uses the aforementioned rate distortion theory coupled with a multivariate Gaussian assumption to provide an additional penalty based on cross-correlation, i.e., minimize the distances between pairs of points belonging to different domains but the same label. The authors demonstrate, that under suitable assumptions, this approach will recover the invariant feature representation. The authors construct several hand-crafted problems where indeed this assumption (and the proposed algorithm) work well, and also provide some experiments on MNIST-type data.

**Summary Of The Review:**

In summary, I recommend a weak reject. I am quite familiar with the related work in this area and I believe that the authors should put in more effort to identify where their approach fails in the real-world and subsequently see how they can relax assumptions made in the paper by analyzing these failure cases. Domain generalization research has seen a rapid increase in methodological work that propose various lenses to look at the problem, however, it is hard to disambiguate and see the value of these approaches when run in the real-world.

---

> ### Author Response · Authors · 2021-11-22
> **Reply to reviewer eG9q**
>
> We thank the reviewer for the feedback. We address their concerns below:
>
> - **cross-domain covariance** Our covariance penalty is not cross-domain covariance. In the Section 5.2, we mention that CORAL (Sun& Saenko, 2016)  "aligns the correlation matrix from different domains", which is one of the typical cross-domain covariance methods. However, our method "decorrelates each dimension of the representation by minimizing the off-diagonal term of cross-correlation matrix", which means this is the cross-correlation between different dimensions of the representation. In our penalty, we do not want to align the covariance between different domains. For example, given a batch of data of size $B\times N$, where $B$ is the batch size and $N$ is the feature dimension. Cross-domain covariance tries to deal with the row vector and the correlation matrix is of size $N\times N$. Our covariance penalty deals with column vector, and the correlation matrix is of size $B\times B$. We  appreciate  references provided by the reviewer, and we have added them into the paper in Section 5.2.
>
> - **Oversimplification of Gaussian** We assume the Gaussian distribution here only to give the rate distortion loss a simpler form and thus facilitate the computations.  Let us note that the recent works of [1, 2] that obtain state-of-the-art results in self-supervised learning also utilize the Gaussian assumption. In order to illustrate that minimizing the loss of (17) will lead to the minimization of rate distortion loss (4). We directly estimate (4) via k-nearest-neighbor (kNN) estimator and report the estimated value of (4) for every checkpoint.  We generally found that using Gaussian penalty will also efficiently reduce the kNN estimated penalty. We add this explanation in the Appendix C.
>
>   We use Twins-Direct to denote the method that uses  kNN estimation directly as penalty. We add the results for MNIST type datasets in Table 2. We can see that Twins-Direct achieve similar performance with the Gaussian based Twins method.
>
>  - **Real-world Datasets** Our work extends a previously published theoretical work on domain generalization. To achieve that, we introduce a link with rate distortion theory and we showcase how our proposal can result in an extension in various experiments. We agree that obtaining results in real-world datasets would strengthen the proposed method, but this is primarily a theoretical work. We have updated the conclusion to mention such large scale experiments with real-world datasets as a future step.
>
>   In addition, we have conducted experiments on PACS and OfficeHome in Section 4.3.
>
>   OfficeHome:
>
>   | Algorithm | A                  | C                  | P                  | R                  | Avg                 |
>   | --------- | ------------------ | ------------------ | ------------------ | ------------------ | ------------------- |
>   | ERM       | 61.3 $\pm$ 0.7     | 52.4 $\pm$ 0.3     | 75.8 $\pm$ 0.1     | 76.6 $\pm$ 0.3     | 66.5  $\pm$ 0.3     |
>   | IRM       | 58.9 $\pm$ 2.3     | 52.2 $\pm$ 1.6     | 72.1 $\pm$ 2.9     | 74.0 $\pm$ 2.5     | 64.3   $\pm$ 2.1    |
>   | Twins-IRM | **64.8 $\pm$ 0.2** | **52.6 $\pm$ 0.7** | **77.5 $\pm$ 0.2** | **78.9 $\pm$ 0.3** | **68.5  $\pm$ 0.4** |
>
>   PACS:
>
>   | Algorithm | A                  | C                  | P                  | S                  | Avg                |
>   | --------- | ------------------ | ------------------ | ------------------ | ------------------ | ------------------ |
>   | ERM       | 84.7 $\pm$ 0.4     | **80.8 $\pm$ 0.6** | 97.2 $\pm$ 0.3     | 79.3 $\pm$ 1.0     | 85.5 $\pm$ 0.7     |
>   | IRM       | 84.8 $\pm$ 1.3     | 76.4 $\pm$ 1.1     | 96.7 $\pm$ 0.6     | 76.1 $\pm$ 1.0     | 83.5 $\pm$ 1.1     |
>   | Twins-IRM | **88.0 $\pm$ 0.3** | 79.6 $\pm$ 0.4     | **97.9 $\pm$ 0.5** | **80.1 $\pm$ 0.9** | **86.4 $\pm$ 0.6** |
>
>
>   [1] Barlow twins: Self-supervised learning via redundancy reduction
>
>   [2] VICReg: Variance-Invariance-Covariance Regularization for Self-Supervised Learning

---

> > ### Author Response · Authors · 2021-11-29
> > **Questions from the reviewer eG9q?**
> >
> > We appreciate the time and effort of the reviewer so far. Given the upcoming deadline for the ending of the discussion period, we are wondering whether our responses with the added experiments and our responses have addressed the concerns of the reviewer.

---

### Author Response · Authors · 2021-11-22
**Common Response**

We are thankful to the reviewers for their feedback. We have performed a number of extensions that we summarize below:

1. We add experiments of Twins-IRM on OfficeHome and PACS in Section 4.3. In addition, we are conducting new experiments during the next week, and we will post the results as soon as we have them.

2. We  improve our theorem so that it does not rely on the domain-free latent $X$. Instead, the proxy from another domain $X^{e'}$ would suffice, which theoretically ensures using instances from another domain to construct contrastive batch. See Theorem 3.2.

3. We use kNN based entropy estimation to directly estimate the IB loss (4) without the Gaussian assumptioin. See Table 2 and Appendix C.

4. We add references to non invariant feature learning methods in Section 1.

5. We rephrase some parts of the paper to make things clearer: Section 5.2/5.3, Assumption 1/4.

In addition to these changes, we answer each reviewer's questions individually below. We kindly ask all reviewers to re-assess our work, if their opinion of our paper has improved after the revision.

---

### Decision · Program_Chairs · 2022-01-20

**Decision:**

Reject

**Comment:**

Casting domain generalization as a rate-distortion problem and developing an information-theoretic approach to solving it looks like an interesting idea. While the proposed method is technically sound, the assumption made in the proposed method is too strong to hold in real-world applications. Though in the rebuttal the authors provided additional experiments on two benchmark datasets, reviewers' concerns about the strong assumption made in the proposed algorithm still remain. To address this issue, I think besides conducting more extensive experiments, the authors also need to analyze when the assumption does not hold in practice, why the proposed algorithm could still perform well compared with other domain generalization methods.

In summary, this is a borderline paper below the acceptance bar of ICLR.